# Does Short-Term Combined Irrigation Using Brackish-Reclaimed Water Cause the Risk of Soil Secondary Salinization?

**DOI:** 10.3390/plants11192552

**Published:** 2022-09-28

**Authors:** Chuncheng Liu, Bingjian Cui, Juan Wang, Chao Hu, Pengfei Huang, Xiaojun Shen, Feng Gao, Zhongyang Li

**Affiliations:** 1Institute of Farmland Irrigation, Chinese Academy of Agricultural Sciences, Xinxiang 453002, China; 2Key Laboratory of High-Efficient and Safe Utilization of Agriculture Water Resources, Chinese Academy of Agricultural Sciences, Xinxiang 453002, China; 3Graduate School of Chinese Academy of Agricultural Sciences, Beijing 100081, China; 4Agriculture Water and Soil Environmental Field Science Research Station of Xinxiang City of Henan Province of Chinese Academy of Agricultural Sciences, Xinxiang 453000, China; 5College of Hydraulic Science and Engineering, Yangzhou University, Yangzhou 225109, China; 6College of Water Conservancy Engineering, Tianjin Agricultural University, Tianjin 300392, China

**Keywords:** mixed irrigation, rotational irrigation, exchangeable sodium percentage (ESP), sodium adsorption ratio (SAR)

## Abstract

Brackish water has to be used to irrigate crops for harvest due to the scarcity of freshwater resources. However, brackish water irrigation may cause secondary soil salinization. Whether the combined utilization of different non-conventional water resources could relieve the risk of secondary soil salinization has not been reported. In order to explore the safe and rational utilization of brackish water in areas where freshwater resources are scarce, a pot experiment was conducted to study the risk of secondary soil mixed irrigation and rotational irrigation using brackish water and reclaimed water or freshwater. The results indicated that: (1) Short-term irrigation using reclaimed water did not cause secondary soil salinization, although increasing soil pH value, ESP, and SAR. The indices did not exceed the threshold of soil salinization. (2) Compared with mixed irrigation using brackish–freshwater, the contents of soil exchangeable Ca^2+^, K^+^, and Mg^2+^ increased, and the content of soil exchangeable Na^+^ decreased under rotational irrigation using brackish-reclaimed water. In addition, the contents of soil exchangeable Na^+^ and Mg^2+^ under mixed irrigation or rotational irrigation were significantly lower, and the exchangeable K^+^ content of the soil was higher compared with brackish water irrigation. The exchangeable Ca^2+^ content under rotational irrigation was higher than that of brackish water irrigation, while the reverse was seen under mixed irrigation. (3) For different combined utilization modes of brackish water and reclaimed water, the ESP and SAR were the lowest under rotational irrigation, followed by mixed irrigation and brackish water irrigation. The ESP under brackish water treatment exceeded 15%, indicating a certain risk of salinization, while ESPs under other treatments were below 15%. Under mixed irrigation or rational irrigation using reclaimed-brackish water, the higher the proportion or rotational times of reclaimed water, the lower the risk of secondary soil salinization. Therefore, short-term combined irrigation using brackish water and reclaimed water will not cause the risk of secondary soil salinization, but further experiments need to verify or cooperate with other agronomic measures in long-term utilization.

## 1. Introduction

Increasing population and decreasing arable land are two major threats to agricultural sustainability [1]. Early in the 21st century, the shortage of global water resources, environmental pollution, and the intensified salinization of soil and water affected agricultural production. Various environmental factors, such as strong winds, extreme temperatures, soil salinity, droughts, and floods, affect the production and cultivation of agricultural crops. As one of the most destructive environmental pressures, soil salinity causes an obvious decrease in the area of arable land, crop productivity, and the quality of crops [1,2] and it is the main factor in the agricultural economy in arid areas [3]. High salinity affects about 20% of the arable land and 33% of irrigated farmland worldwide. Salinized areas are increasing at an annual rate of 10% due to various reasons, such as low rainfall, high surface evaporation, primary rock weathering, saline water irrigation, and undesirable farming practices. It is estimated that the salinized arable land will exceed 50% by 2050 [4]. Water scarcity in arid areas is driving irrigated agriculture to use more marginal or inferior water sources [5]. Unconventional water resources are an alternative or supplementary resource, and their rational use in agriculture can alleviate the shortage of freshwater resources. So, the safe utilization of unconventional water resources has received more and more attention. In areas where fresh water is scarce, brackish water has to be used to irrigate crops for the harvest. The salinity in irrigation water is the main factor restricting crop yield under brackish water irrigation. Combined irrigation, using brackish water and reclaimed water instead of freshwater, can alleviate the pressure of shortage of agricultural water resources, increase crop yield and ameliorate the accumulation of salt in soil due to saline water irrigation. Therefore, it is of great significance to study the combined irrigation mode and its effect on secondary soil salinization using brackish water and reclaimed water.

It has been reported that brackish water not only can motivate the growth of crops without an obvious decrease or even an increase in yield but also improve water-use efficiency at the appropriate salinity in water [6]. Brackish water is rich in beneficial micronutrients but may cause toxic stress to plant growth by concentrating certain ions, such as sodium ions, chloride ions, and bicarbonate ions [7,8]. This is due to sodium ions replacing potassium ions in the biochemical reaction and the conformational changes in proteins induced by sodium ions and chlorine ions [9]. Furthermore, soil salinity may disturb the balance of nitrogen, calcium, potassium, phosphorus, iron, zinc, and boron in plants, cause imbalances in nutrition or interfere with nutrient uptake [7,9,10]. Soil salt plays a major role in affecting the fluorescence parameters, yield, and quality of most processing tomatoes compared to nitrogen, and reducing nitrogen could obtain higher yields in areas with high salinity [11,12]. It has been reported that saline water ice performs better than freshwater ice due to it increasing the yield of rice and saving freshwater resources [13]. Soil moisture is an important factor in soil fertility and is the main source of crop water uptake. Salinity is the main index of soil salinization, and the movement of salt is closely related to soil water. Usually, electrical conductivity (EC) is used to represent the soil salt content. In order to avoid secondary salinization, it is recommended that the salinity of brackish water should not exceed 8.8 dS m^−1^ under the drip irrigation of cotton with saline water in the low plain near the Bohai sea in Northern China. Irrigating freshwater (1.2 dS m^−1^) in the growth stage can improve the potential yield, and irrigating saline water of 7 dS/m in the reproductive stage can improve the fruit quality, and moderate brackish water (<4.5 dS m^−1^) can also be used in the whole growth stage. [14]. Based on HYDRUS, twenty-year consecutive irrigation with brackish water (3 g L^−1^) is relatively more suitable in homogeneous soil rather than heterogeneous soil in North China [15]. Due to the low dissolved salt content of brackish water, short-term irrigation with brackish water did not obviously affect the chemical characteristics and salinization of the soil, but long-term irrigation may cause the occurrence of salinization [16] and soil water repellency [17], and huge changes in the physical and chemical properties of the soil [18,19]. Furthermore, improving the grain protein content by using brackish water irrigation has been reported [20]. Therefore, brackish water irrigation may cause secondary soil salinization and enhance the spatial variation in soil water. Theoretically, the salinity in reclaimed water was lower than in saline water or brackish water. Irrigating crops with reclaimed water can leach salt, and mixing with brackish water can dilute each other, reducing the salinity of the brackish water, thus avoiding secondary salinization. However, it has not been reported whether leaching salt or reducing the salinity has better effects on avoiding secondary salinization. The existing research on reclaimed water utilization mainly involves suitable irrigation technology for reclaimed water [21] and the influences of reclaimed water irrigation on crop growth [22], quality [23], soil environment [24], soil microbial community structure [25], groundwater [26], etc. Romero-Trigueros et al. verified the feasibility of medium-term or long-term reclaimed water irrigation in citrus [27]. For example, the concentration of micronutrients in the leaves did not exceed the threshold [28] after 15-year irrigation with reclaimed water. It is feasible to use reclaimed water for irrigation due to the lack of impact on the concentrations of heavy metals and micronutrients in the leaves and fruits, such as B, Na, and Zn [29]. Therefore, in this study, different mixed irrigation and rotational irrigation using brackish water and reclaimed water were set up to explore the influences on secondary soil salinization through pot experiments in order to provide a theoretical basis for the safe utilization of brackish water and reclaimed water.

## 2. Results

### 2.1. Soil Water and Salt Contents

The variations in the soil water content and EC under different irrigation scenarios with brackish water and reclaimed water or freshwater after crop harvesting are shown in Figure 1.

As shown in Figure 1a, the soil water content under reclaimed water irrigation (R) increased by 1.10% without a significant difference compared with freshwater irrigation (F). Compared with the mixed irrigation using brackish water and freshwater, the soil moisture content was slightly higher at a 1:1 mixed ratio and lower at a 1:2 mixed ratio under mixed irrigation using brackish water and reclaimed water, but there was no significant difference between the two treatments. Under mixed irrigation using brackish water and reclaimed water, the soil moisture content decreased gradually with the increase in the proportion of reclaimed water in the mixture. Therefore, there was no significant difference in the soil moisture content between reclaimed water irrigation and freshwater irrigation, and reclaimed could be as freshwater to mixed irrigation with brackish water.

Compared with freshwater irrigation (F), the soil EC was 0.818 dS m^−1^ under reclaimed water irrigation (R), which significantly increased by 49.6%. Under mixed irrigation using brackish–freshwater, the soil EC decreased slightly at a 1:1 mixed ratio and increased at a 1:2 mixed ratio compared with mixed irrigation using brackish-reclaimed water, but there were both no significant differences. There was an opposite change trend with soil water. Under mixed irrigation using brackish–reclaimed water, the soil EC decreased gradually with the increase in the proportion of reclaimed water in the mixture, and there was no significant difference between a 1:1 and a 1:2 mixed irrigation. Therefore, the soil salinity is mainly determined by the salt content in irrigation water. This is consistent with the changing trend of the soil moisture content under mixed irrigation using brackish–reclaimed water, and the reason may be that the higher the salt content, the stronger the limiting effect on crop water uptake, and more water remaining in the soil.

As seen in Figure 1b, compared with rotational irrigation using freshwater and brackish water, the soil water content was slightly higher under rational irrigation using reclaimed water and brackish water, but there was no significant difference on the whole; however, the soil EC increased significantly on the whole (*p* < 0.05), with an increase of 4.97~18.35%. Under rotational irrigation using reclaimed water and brackish water, the soil moisture content and EC decreased gradually with the increase in irrigation times of reclaimed water, and the soil moisture content and EC under the rotational irrigation treatment were significantly lower than those under pure brackish water irrigation (*p* < 0.05). Therefore, rotational irrigation using reclaimed water and brackish water had no significant effect on soil moisture but significantly impacted soil salinity. There was a negative relationship between the rotational times of the reclaimed water and the soil salt content.

Comparing mixed irrigation and rotational irrigation using brackish water and reclaimed water (Figure 1c), the soil moisture content under the “reclaimed–brackish water” rotational irrigation treatment (RR1) was slightly lower than that under a 1:1 mixed irrigation treatment (MR1), while the soil water content under the “reclaimed-reclaimed–brackish water” rotation irrigation treatment (RR2) was slightly higher than that of the 1:2 mixed irrigation treatment (MR2). However, there was no significant difference at a level of 0.05 between mixed irrigation and rotational irrigation. In addition, regardless of rotational irrigation or mixed irrigation using reclaimed water and brackish water, the soil moisture content was significantly lower compared to pure brackish water irrigation (*p* < 0.05). Similar to the soil water content, the soil EC under the rotational or mixed irrigation was also significantly lower than that under pure brackish water irrigation. The soil EC under rotational irrigation was slightly higher compared to mixed irrigation, but there was no significant difference between treatments.

### 2.2. Soil Exchangeable Ions Contents

Soil exchangeable capacity can adjust the soil solution concentration, ensure the diversity of solution components, and maintain “physiological balance”, which is of great significance for plant nutrition and fertilization. The changes in soil exchangeable ion content after harvesting under different irrigation scenarios using brackish water and reclaimed water or freshwater are shown in Figure 2.

As seen from Figure 2a, compared with freshwater irrigation (F), the contents of soil exchangeable Ca^2+^, Na^+^, K^+^, and Mg^2+^ all decreased under reclaimed water irrigation (R). Between R and F, the differences in the content of soil exchangeable Ca^2+^ and Mg^2+^ reached a significant level (*p* < 0.05) with decreases of 34.06% and 19.09%, respectively. Compared with mixed irrigation using brackish water and freshwater, the contents of soil exchangeable Ca^2+^ and Mg^2+^ decreased, but the difference was not significant, while the contents of soil exchangeable Na^+^ and K^+^ increased, with the former reaching a significant level (*p* < 0.05) under mixed irrigation using brackish water and reclaimed water. It can be seen that using reclaimed water instead of fresh water to mix with brackish water will cause an increase in soil exchangeable Na^+^ content and a reduction in the contents of exchangeable Ca^2+^ and Mg^2+^.

For mixed irrigation using brackish water and reclaimed water, with the increase in the proportion of reclaimed water in the mixture, the content of soil exchangeable Na^+^ decreased significantly (*p* < 0.05), while there were no significant differences in the contents of soil exchangeable Ca^2+^, K^+^ and Mg^2+^ between treatments. So, mixed irrigation with brackish water and reclaimed water was more effective in reducing the soil exchangeable Na^+^ content than pure brackish water irrigation.

As shown in Figure 2b, compared with rotational irrigation using freshwater and brackish water, except that the RR1 treatment was slightly lower than the RF1 treatment, the soil exchangeable Ca^2+^, Na^+^, K^+^, and Mg^2+^ contents increased under rotational irrigation using reclaimed water and brackish water, of which the soil exchangeable Na^+^ content significantly increased, and the soil exchangeable K^+^ content in RR2 was significantly higher than that in RF2. Therefore, using reclaimed water instead of freshwater to irrigate with brackish water rotationally will increase the soil’s exchangeable ion content, especially the exchangeable Na^+^ content.

Under rotational irrigation using reclaimed water and brackish water, with the increase in irrigation times of reclaimed water, soil exchangeable Ca^2+^ and K^+^ contents increased and then decreased under pure reclaimed water irrigation to some extent. There was no significant difference in exchangeable Ca^2+^ between the treatments, and the exchangeable K^+^ content in the RR2 treatment was significantly higher compared to the B treatment. The soil exchangeable Na^+^ content showed a decreasing trend with the increase in the irrigation times of reclaimed water, and there was a significant difference between the rotational irrigation treatment and the pure brackish water irrigation or the reclaimed water irrigation treatment. No significant difference in the soil exchangeable Mg^2+^ content was found between the treatments. It can be seen that, compared with pure brackish water irrigation, rotational irrigation had a significant effect on reducing soil exchangeable Na^+^ content but no significant effect on other exchangeable ions contents.

Comparing mixed irrigation and rotational irrigation using brackish water and reclaimed water (Figure 2c), the soil exchangeable Ca^2+^, K^+^, and Mg^2+^ contents showed an increasing trend under rotational irrigation compared to mixed irrigation, of which the exchangeable Ca^2+^ content increased significantly. The exchangeable K^+^ content in RR2 was significantly higher than that in MR2. The soil exchangeable Na^+^ content under rotational irrigation decreased compared to mixed irrigation, of which RT1 was significantly lower than MR1 treatment. In addition, under mixed irrigation or rotational irrigation, (1) the exchangeable Na^+^ content was significantly lower than that under pure brackish water irrigation, (2) the exchangeable K^+^ content was higher than that under pure brackish water irrigation, of which RR2 was significantly higher than B (*p* < 0.05). The exchangeable Ca^2+^ content under rotational irrigation was higher than that under pure brackish water irrigation and vice versa for mixed irrigation. The soil exchangeable Mg^2+^ contents under rotational irrigation or mixed irrigation had no significant difference from reclaimed water irrigation and were both lower than that under pure brackish water irrigation, of which the soil exchangeable Mg^2+^ content in MR1 was significantly lower than B (*p* < 0.05).

### 2.3. pH in Soil

Soil pH is closely related to microbial activities, synthesis, the decomposition of organic matter, the ability to retain nutrients, and the migration of elements in the process of soil occurrence. The changes in the soil pH value under different irrigation scenarios are shown in Figure 3.

Figure 3a shows that the soil pH value in R was 7.91, which was 1.54% higher than that in F, and the difference was significant (*p* < 0.05) between the two treatments. Compared with mixed irrigation using brackish water and freshwater, the soil pH value showed an increasing trend, and the difference reached a significant level (*p* < 0.05) under mixed irrigation using brackish water and reclaimed water. Under mixed irrigation using brackish water and reclaimed water, with the increase in the proportion of reclaimed water in the mixture, the soil pH values in the mixed irrigation treatments were higher than that in pure brackish water irrigation, and the difference was significant (*p* < 0.05).

As seen in Figure 3b, compared with rotational irrigation using freshwater and brackish water, the soil pH value increased without significant difference under rotational irrigation using reclaimed water and brackish water. For the rotational irrigation using reclaimed water and brackish water, the soil pH value under rotational irrigation was significantly higher (*p* < 0.05) than in pure brackish water irrigation. On the whole, the pH values in all treatments did not exceed 8.5, indicating that there was no risk of soil alkalization.

As shown in Figure 3b, compared to mixed irrigation using reclaimed water and reclaimed water, the soil pH value was significantly lower, with a decrease of 0.38–0.63% under rotational irrigation using reclaimed water and brackish water. In addition, the soil pH value under mixed irrigation and rotational irrigation were both significantly higher than those under pure brackish water irrigation, but the pH values were less than 8.5 without the risk of soil alkalization.

### 2.4. Exchangeable K^+^/Na^+^ Content in Soil

The higher the soil exchangeability K^+^/Na^+^, the lower the soil salinization hazard. Figure 4 shows the changes in soil exchangeable K^+^/Na^+^ after harvesting under different irrigation scenarios.

As shown in Figure 4a, the soil exchangeable K^+^/Na^+^ in R was 0.51, which was 5.56% lower than that in F, but the difference was not significant. The soil exchangeable K^+^/Na^+^ under mixed irrigation using brackish water and freshwater showed a decreasing trend compared to mixed irrigation using brackish water and reclaimed water. Under mixed irrigation of brackish water and reclaimed water, the soil exchangeable K^+^/Na^+^ increased significantly (*p* < 0.05) with the increase in the proportion of reclaimed water in the mixture.

Figure 4b shows that the soil exchangeable K^+^/Na^+^ decreased obviously after using reclaimed water instead of freshwater to irrigate the crops rotationally with brackish water. Under rotational irrigation using reclaimed water and brackish water, soil exchangeable K^+^/Na^+^ showed an increasing trend with the increase in irrigation times of reclaimed water, and the difference was significant between treatments.

As seen in Figure 4c, compared to mixed irrigation using reclaimed water and reclaimed water, the soil exchangeable K^+^/Na^+^ was obviously higher on the whole under rotational irrigation using reclaimed water and brackish water. In addition, the soil exchangeable K^+^/Na^+^ under mixed irrigation and rotational irrigation were both significantly higher than those under pure brackish water irrigation.

### 2.5. ESP and SAR

ESP and SAR are two main indicators to evaluate the risk of secondary soil salinization. The changes in soil ESP and SAR after harvesting under different irrigation scenarios are shown in Figure 5.

As shown in Figure 5a, the ESP and SAR in R were 5.42% and 0.69, which significantly increased by 46.31% and 362% compared with MC1, respectively. However, the ESP and SAR were far below the threshold of soil salinization (15% and 13 (mmol L^−1^)^0.5^). Compared with mixed irrigation using brackish water and freshwater, the soil ESP and SAR showed an increasing trend, and the difference was significant (*p* < 0.05) under mixed irrigation using brackish water and reclaimed water. Under mixed irrigation using brackish water and reclaimed water, soil ESP and SAR decreased significantly with the increase in the proportion of reclaimed water in the mixture, indicating that mixed irrigation had an obvious effect on reducing soil ESP and SAR.

Figure 5b shows that, compared with rotational irrigation using freshwater and brackish water, the soil ESP and SAR increased significantly (*p* < 0.05) under rotational irrigation using reclaimed water and brackish water. With the increase in the irrigation times of reclaimed water, soil ESP and SAR had an obvious decrease (*p* < 0.05) for rotational irrigation using reclaimed water and brackish water.

For different combined irrigation modes using reclaimed water and brackish water (Figure 5c), ESP and SAR under rotational irrigation were lower than those under mixed irrigation, and the difference in ESP reached a significant level of 0.05 between the two irrigation modes. There was a significant difference in SAR between MR1 and RR1, while there was no significant difference in SAR between MR2 and RR2. However, regardless of rotational or mixed irrigation, neither soil ESP nor SAR exceeded 15% and 13 (mmol L^−1^)^0.5^, and they were significantly lower than those under pure brackish water irrigation.

### 2.6. Relationship between ESP and SAR

ESP and SAR represent the state of sodium in the soil. The change curve between the soil ESP and SAR after harvest under different combinations of brackish water and reclaimed water irrigation are shown in Figure 6. As shown, there was a good correlation between the soil ESP and SAR, with the Pearson’s correlation coefficient above 0.967. Through the linear fitting for soil ESP and SAR, there was a good linear relationship between them, and the fitting formula was that ESP = 6.95302SAR + 1.35353, R^2^ = 0.92865.

### 2.7. Crop Biomass

Figure 7a illustrates that the AFW and ADW in R increased by 7.07% and 5.25% compared to F, respectively, but the difference did not reach a significant level. AFW and ADW under mixed irrigation with reclaimed water and brackish water decreased slightly compared with those under mixed irrigation with freshwater and brackish water. Under mixed irrigation with reclaimed water and brackish water, there were no differences in AFW and ADW between treatments with an increase in the proportion of reclaimed water in the mixture.

As seen in Figure 7b, compared with rotational irrigation with freshwater and brackish water, AFW and ADW increased slightly under rotational irrigation with freshwater and brackish water. Under rotational irrigation with reclaimed water and brackish water, AFW and ADW under the rotational irrigation treatments were higher than those under pure brackish water irrigation, and the top occurred in pure reclaimed water irrigation, followed by RR1.

For different irrigation modes (Figure 7c), there were no differences in AFW and ADW between rotational irrigation and mixed irrigation, except that ADW in RR1 was significantly higher than that in MR1. In addition, AFW and ADW, whether rotational irrigation or mixed irrigation, were improved compared with pure brackish water irrigation, of which RR1 was significantly higher than B.

### 2.8. Leaf Sodium Content

Table 1 shows the leaf sodium content under different irrigation modes. The leaf sodium content was the highest under pure brackish water irrigation and the lowest under pure reclaimed water irrigation. Leaf sodium content in B was significantly higher than those in other treatments except MR1. For different irrigation modes, there was a significant difference between MR1 and RR1 but no difference between MR2 and RR2.

## 3. Discussion

### 3.1. Response of Soil Exchangeable Ions Contents to Combined Irrigation Modes Using Brackish-Reclaimed Water

Soil exchangeability is the basis of nutrient availability for plants and microorganisms [30]. Soil exchange capacity can adjust the concentration of soil solution to maintain “physiological balance” and maintain nutrients from being leached by rain. Exchangeable base ions are important indicators of soil nutrients and can represent soil fertility to some extent. Ca^2+^, Mg^2+^, and K^+^ are the main components of soil exchangeable base ions, and the interaction between their exchange capacity and nutrients (N, P, etc.) is an important chemical index in the soil to maintain the health and stability of a terrestrial ecosystem [31]. The results in this paper showed that the contents of exchangeable Ca^2+^, K^+^, and Mg^2+^ in soil under rotational irrigation using reclaimed water and brackish water showed an increasing trend compared to mixed irrigation, in which the content of exchangeable Ca^2+^ increased significantly, indicating that rotational irrigation was conducive to plant growth because exchangeable Ca^2+^, K^+^, and Mg^2+^ were essential nutrients for plant growth and their exchange capacity was significantly related to plant absorption and utilization, which could reflect the bioavailability of soil nutrients [32]. The soil exchangeable Na^+^ content under rotational irrigation using reclaimed water and brackish water is lower than that of mixed irrigation; the reason may be that the high concentration of potassium under rotational irrigation inhibited the adsorption of sodium ions by the soil [33]. The low exchangeable Na^+^ content was not readily absorbed by the crops [34], for example, the Na^+^ contents in the leaves (12.36, 12.43 mg g^−1^) under the rotational irrigation treatment (RR1, RR2) were lower than those (14.46, 12.91 mg g^−1^) under the mixed irrigation treatments (MR1, MR2). This paper also indicated that the content of soil exchangeable Na^+^ under mixed irrigation or rotational irrigation was significantly lower than that under pure brackish water irrigation and significantly higher than that in reclaimed water irrigation, which was mainly determined by the content of Na^+^ in different water sources. The exchangeable K^+^ content under rotational irrigation or mixed irrigation was higher than that under pure brackish water irrigation, while the exchangeable Mg^2+^ content under rotational irrigation was lower than that under pure brackish water irrigation. The opposite change trend in exchangeable Mg^2+^ and K^+^ may be caused by the antagonistic function between K^+^ and Mg^2+^, competing for the adsorption sites [33]. The exchangeable Ca^2+^ content under rotational irrigation was higher than that under pure brackish water irrigation, but there was the opposite trend for mixed irrigation, indicating that rotation irrigation was conducive to the maintenance of soil exchangeable Ca^2+^. Now, few studies about the combined utilization of reclaimed water and brackish have been reported. Our research could provide new perspectives to solve the existing problems of brackish water or reclaimed water. Our research is a continuation or expansion of the single utilization of reclaimed or brackish water. However, we also have many problems that need to be solved, such as the effects of long-term irrigation on soil, crop, and soil microorganisms, as well as the interaction mechanism of ions.

### 3.2. Response of Secondary Soil Salinization to Combined Irrigation Modes Using Brackish Water and Reclaimed Water

The soil pH and ESP are usually used to determine alkalized soil at home and abroad [35] and are also the main factors of soil dispersion [36]. Generally, the pH was above 8.5, and the ESP exceeded 15% in alkaline soil. Through a 15-year experiment under reclaimed water irrigation, Tahtouh et al. found that the physical and chemical properties of the soil were not significantly affected, and SAR and ESP were within the threshold [16]. Guedes et al. also found that the physical and chemical properties and functions of soil were not affected by short-term reclaimed water irrigation [37]. Therefore, the utilization of reclaimed water was feasible [29]. In this paper, the results showed that pH, ESP, and SAR under reclaimed water irrigation were significantly higher than those under freshwater irrigation. However, the pH did not exceed 8.5, and ESP and SAR were also far lower than the threshold of soil salinization (15% and 13 (mmol L^−1^)^0.5^) under reclaimed water irrigation, so short-term reclaimed water irrigation would not cause secondary soil salinization.

The salinity of the irrigation water did not necessarily result in secondary soil salinization, but some tillage measures should be applied to prevent the long-term utilization of saline water [38]. The experimental results in this paper showed that the soil ESP under brackish water irrigation was slightly more than 15%, indicating a possible risk of secondary soil salinization. However, whether mixed irrigation or rotational irrigation using brackish water and reclaimed water or freshwater, the soil ESPs were all significantly reduced compared with pure brackish water irrigation. In addition, compared with mixed irrigation or rotational irrigation using freshwater and brackish water, the soil ESP and SAR were significantly increased after freshwater being replacing with reclaimed water. However, ESP and SAR were all within the threshold, indicating that short-term mixed irrigation or rotational irrigation using freshwater/reclaimed water and brackish water could not cause secondary soil salinization. Therefore, the short-term combined utilization of reclaimed water and brackish water could be applied to alleviate the potential risk of direct brackish water irrigation where freshwater is scarce.

Our results showed that under different irrigation modes of brackish water and reclaimed water, the ESP and SAR in the rotational irrigation treatment were lower than those in the mixed irrigation treatment, but the ESP and SAR were both within the threshold of salinization (15% and 13 (mmol L^−1^)^0.5^). However, the ESP and SAR under the above two irrigation modes were significantly lower than that under pure brackish water irrigation. Due to the SAR calculated in this paper based on the ion content of a 1:5 soil-to-water extract, it was not suitable to judge soil alkalization according to the threshold of SAR. Because the threshold of SAR was calculated by the ion content of saturated mud extract. Therefore, according to the correlation between ESP and SAR (ESP = 6.95302SAR + 1.35353, R^2^ = 0.92865), when ESP was 15%, the corresponding SAR should be 1.96. It was worth noting that the fitting relationship between ESP and SAR was different under different soil textures, salinity, different regions, and other conditions. So, the fitting curve was only applicable to the soil types in the experiment.

## 4. Materials and Methods

### 4.1. Tested Soil

The tested soil was collected from the topsoil (0–20 cm) in a field near the Agricultural Soil and Water Environment Field Scientific Observation and Experiment Station of Xinxiang of the Chinese Academy of Agricultural Sciences. The soil was air-dried, crushed, and sieved (2 mm). The bulk density of the soil was 1.40 g cm^−3^, the field water-hold capacity of the soil was 0.17 g g^−1^, the total nitrogen content was 0.385 g kg^−1^, the total phosphorus content was 0.668 g kg^−1^, the electrical conductivity of the 1:5 soil–water extract was 0.264 dS m^−1^, and the organic matter content was 2.31 percent. Based on the BT-9300HT laser particle sizer (Bettersize Instruments Ltd., Dandong, China), clay particles (<0.002 mm), silt particles (0.002~0.02 mm), and sand particles (0.02~2 mm) accounted for 20.90%, 44.62%, and 34.48%, respectively, and the soil had a loam texture.

### 4.2. Experimental Device and Scheme

The pot experiment was performed from October to December 2020 in the greenhouses of the Agriculture Water and Soil Environmental Field Science Research Station of Xinxiang of the Chinese Academy of Agricultural Sciences. The station is located at 35°19′ N, 113°53′ E, 73 m above sea level, with an average annual temperature of 14.1 °C and multiyear average annual precipitation and evaporation of 588 mm and 2000 mm, respectively. The frost-free period lasts for 210 d, and the average annual sunshine duration is 2398 h.

The tested crop is Shanghai green, also known as cabbage, green cabbage, etc., and is the main fast-growing vegetable cultivated. The experiment set two kinds of irrigation methods, including mixed irrigation and rotational irrigation. There were four levels under mixed irrigation using brackish water and reclaimed water, and with mixed irrigation using brackish–freshwater as the control group. Four levels were also set under rational irrigation using brackish water and reclaimed water, with rotational irrigation using brackish–freshwater as the control group. According to the prior results [34], water with salinity levels of 2–5 g L^−1^ accounts for 47.8% of this saline groundwater in the low plain around the Bohai Sea. Therefore, we selected 3 g L^−1^ of brackish water. Eleven treatments were used with 3 replicates for each treatment in the experiment. The specific experimental design is shown in Table 2. The pot had an upper diameter of 25 cm, a lower diameter of 14.5 cm, a height of 19 cm, and three holes in the bottom. Each pot was loaded with 7 kg of soil (about 17 cm in height) according to the actual bulk density in the field, and 1 g of compound fertilizers (N-P_2_O_5_-K_2_O ratio of 15-15-15) was added to 1 kg of soil according to the local conventional fertilization rate. All of the treatments received fertilizer as a basal application and were irrigated with freshwater before sowing to maintain moisture. The seeds were sown on 9 October 2020 and spread evenly in each pot. Five seedlings were left in each pot at the two-leaf stage (31 October) to start the irrigation treatments. The traditional surface irrigation method was used to irrigate the crops when the soil moisture was lower than 75% of the field water-holding capacity, and the upper limit was 100% of the field water-holding capacity. After the irrigation treatments, the crops were irrigated seven times, and the total irrigation volume was 2.1 L per pot. The soil moisture was monitored using a portable soil moisture meter. There was no drainage during the whole growth stage. The reclaimed water was from the Luotuowan Domestic Sewage Treatment Plant in Xinxiang City, Henan Province. The plant adopted the A^2^/O treatment process, and the water quality was in line with the standard (GB5084-2021). The freshwater source was tap water. The reclaimed water and freshwater had a stable level of components due to their stable sources. The brackish water was obtained by adding sea salt to freshwater according to the existing results from the existing results [39]. The water qualities of different water sources are shown in Table 3.

### 4.3. Measured Indexes and Methods

The soil samples were collected from the pots after the crops were harvested, then they were air-dried at room temperature and then ground to pass through a 2 mm sieve. The oven-drying method was used to measure the soil water content. Part of the soil samples was extracted with a soil-to-water ratio of 1:5, and the extracts were used to determine the conductivity. (EC) was measured using a conductivity meter, water-soluble soil Na^+^ using flame photometry, and the Ca^2+^ and Mg^2+^ contents using EDTA titration [40]. The sodium adsorption ratio (SAR) can be calculated using Equation (1) [41] as follows:(1)$$\mathrm{SAR}=\frac{{\mathrm{Na}}^{+}}{\sqrt{{\mathrm{Ca}}^{2+}+{\mathrm{Mg}}^{2+}}}$$
where the Na^+^, Ca^2+^, and Mg^2+^ concentrations at a soil-to-water ratio of 1:5 are in mmol L^−1^.

The other part of the soil samples was extracted with a soil-to-water ratio of 1:2.5 to measure the pH value using the potentiometric method. The rest of the soil samples were washed with 70% ethanol and exchanged with 0.1 mol L^−1^ ammonium chloride—a 70% ethanol solution. Flame spectrophotometry (Flame Photometer FP6410, Shanghai Xinyi Instrument Co., Ltd., Shanghai, China) was used to measure the exchangeable soil K^+^ and Na^+^ contents, and atomic absorption spectrophotometry (AA7000F, Shimadzu, Kyoto, Japan) was used to measure the exchangeable soil Ca^2+^ and Mg^2+^ contents. The exchangeable soil sodium percentage (ESP) can be calculated using Equation (2) as follows:(2)$$\mathrm{ESP}=\frac{\;\mathrm{exchangeable}\;{\mathrm{Na}}^{+}}{\mathrm{ECEC}}$$
where the exchangeable Na^+^ concentration is in cmol kg^−1^; ECEC refers to the effective action exchange capacity, and can be calculated using Equation (3) as follows:(3)$$\mathrm{ECEC}=\mathrm{exchangeable}\;K^{+}+{\mathrm{Na}}^{+}+{\mathrm{Ca}}^{2+}+{\mathrm{Mg}}^{2+}$$
where all concentrations are in cmol kg^−1^.

After harvesting, the crop samples were collected, rinsed with distilled water, and air-dried. The aboveground fresh weight (AFW) was determined using a balance. Then, the samples were placed in a 105 °C oven for 15 min and then dried to a constant weight at 75 °C to determine the aboveground dry weight (ADW).

### 4.4. Data Analysis

Excel 2010 software was used to calculate the experimental data. The statistical differences among the groups were determined by analysis of variance (ANOVA) using the SPSS25.0 software (IBM Crop., Armonk, NY, USA), followed by the least significant difference (LSD) test for multiple comparisons among groups. A difference returning a *p*-value less than 5% (*p* < 0.05, *n* = 3) was considered statistically significant. The origin2019b software was used to draw the figures.

## 5. Conclusions

(1) Reclaimed water irrigation increased soil pH, ESP, and SAR, but they were all within the threshold of soil salinization. Short-term reclaimed water irrigation did not cause secondary soil salinization.

(2) For different combined utilization modes of brackish water and reclaimed water, the ESP and SAR were the lowest under rotational irrigation, followed by mixed irrigation and brackish water irrigation. In addition, the ESP in brackish water treatment exceeded 15% with a potential risk of soil alkalization, while the ESP in other treatments was all lower than 15% without risk of soil alkalization.

(3) Rotational irrigation or mixed irrigation with reclaimed water and brackish water could improve AFW and ADW compared to pure brackish water irrigation, especially “reclaimed-brackish water” rotational irrigation.

In addition, only the salinity of 3 g L^−1^ in brackish water was considered in this experiment. The higher salinity in brackish water and soils with different textures should be considered to verify and improve the results in this paper. Moreover, for pot experiments, due to the limited height of the pot, it could not reflect the distribution of soil salt in the profile and the leaching effect of salt. So, the experimental results may not agree with long-term experimental results, and experimental field research needs to be carried out in the future.

## Figures and Tables

**Figure 1 plants-11-02552-f001:**
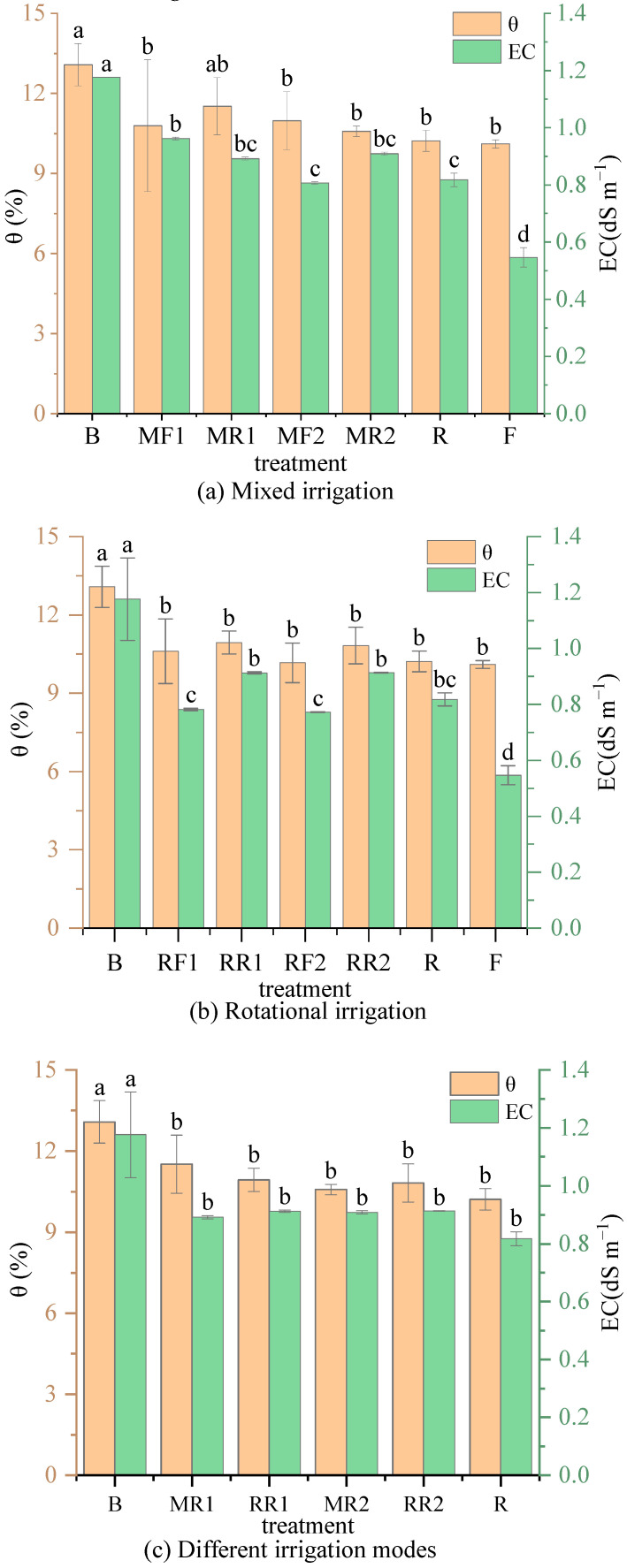
Variations in soil water and salt content under different irrigation scenarios. Note: different lowercase letters on the bars represent the significant differences at the level of 0.05.

**Figure 2 plants-11-02552-f002:**
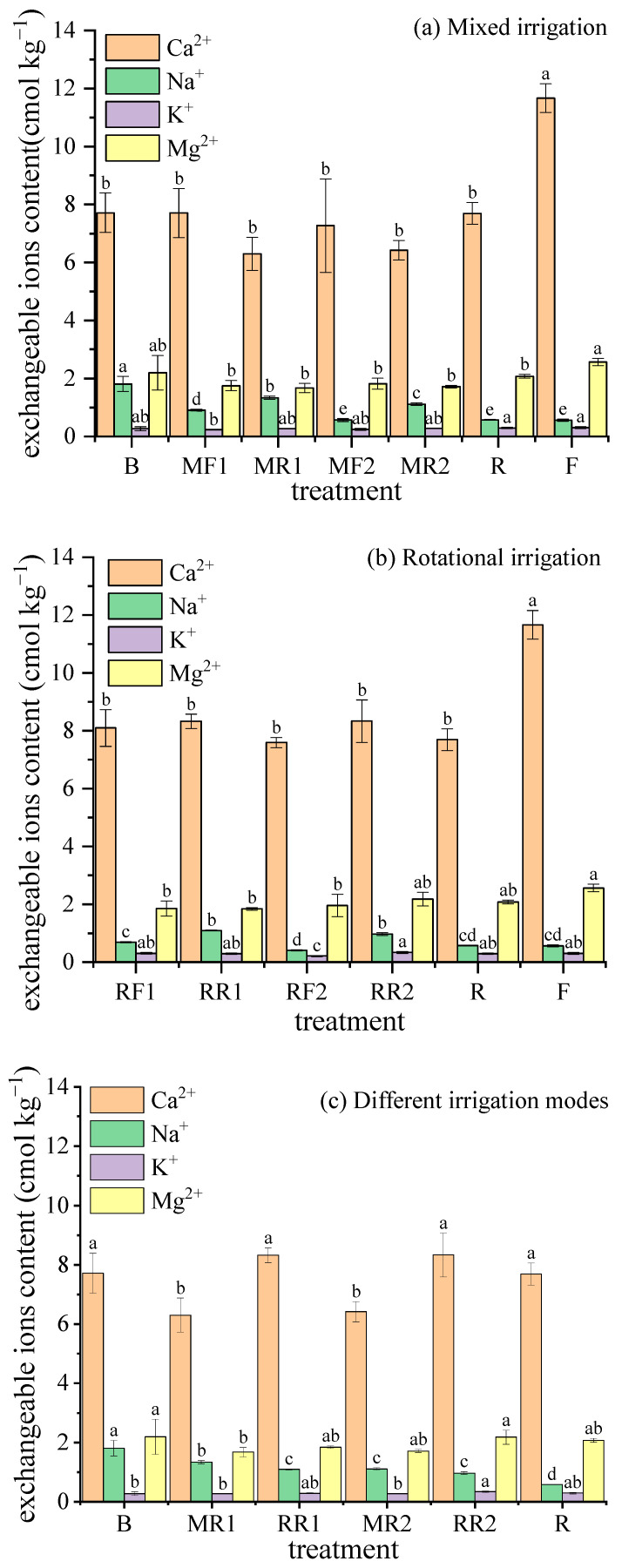
Variations in exchangeable ions contents of soil under different irrigation scenarios. Note: different lowercase letters on the lines represent the significant differences at the level of 0.05.

**Figure 3 plants-11-02552-f003:**
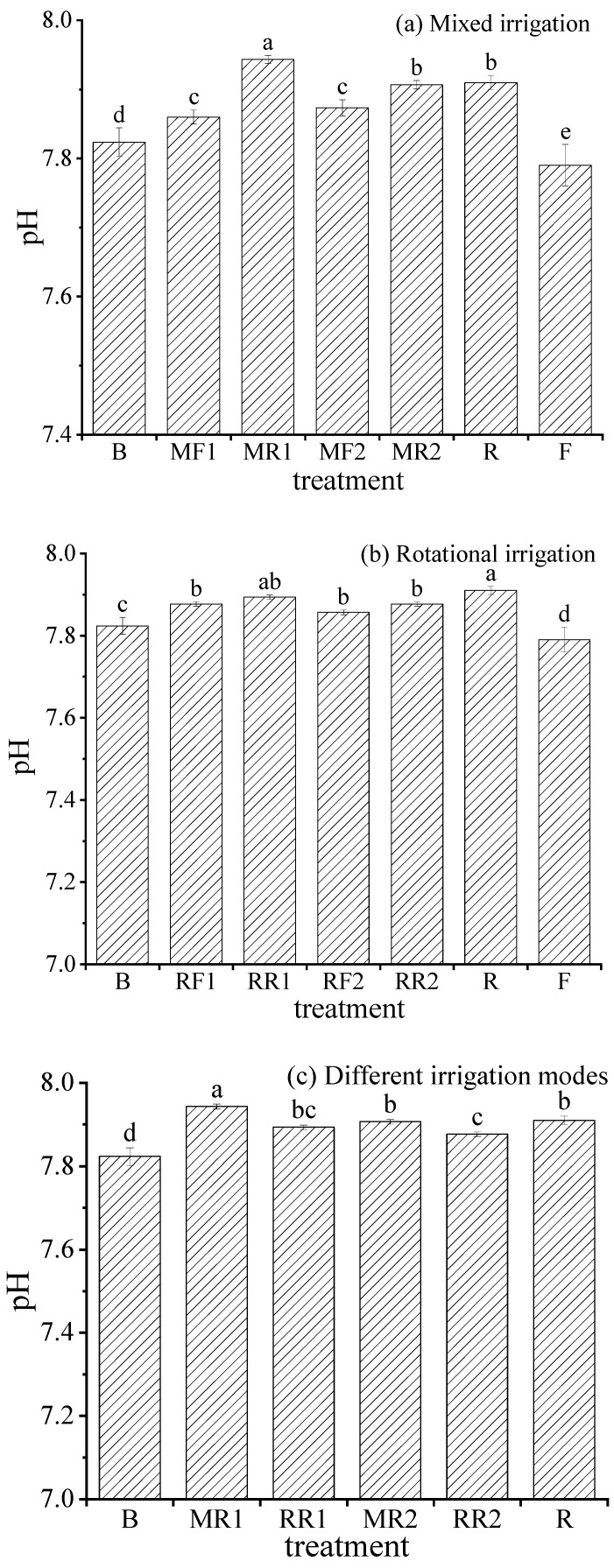
Variation in soil pH value under different irrigation scenarios. Note: different lowercase letters on the bars represent the significant differences at the level of 0.05.

**Figure 4 plants-11-02552-f004:**
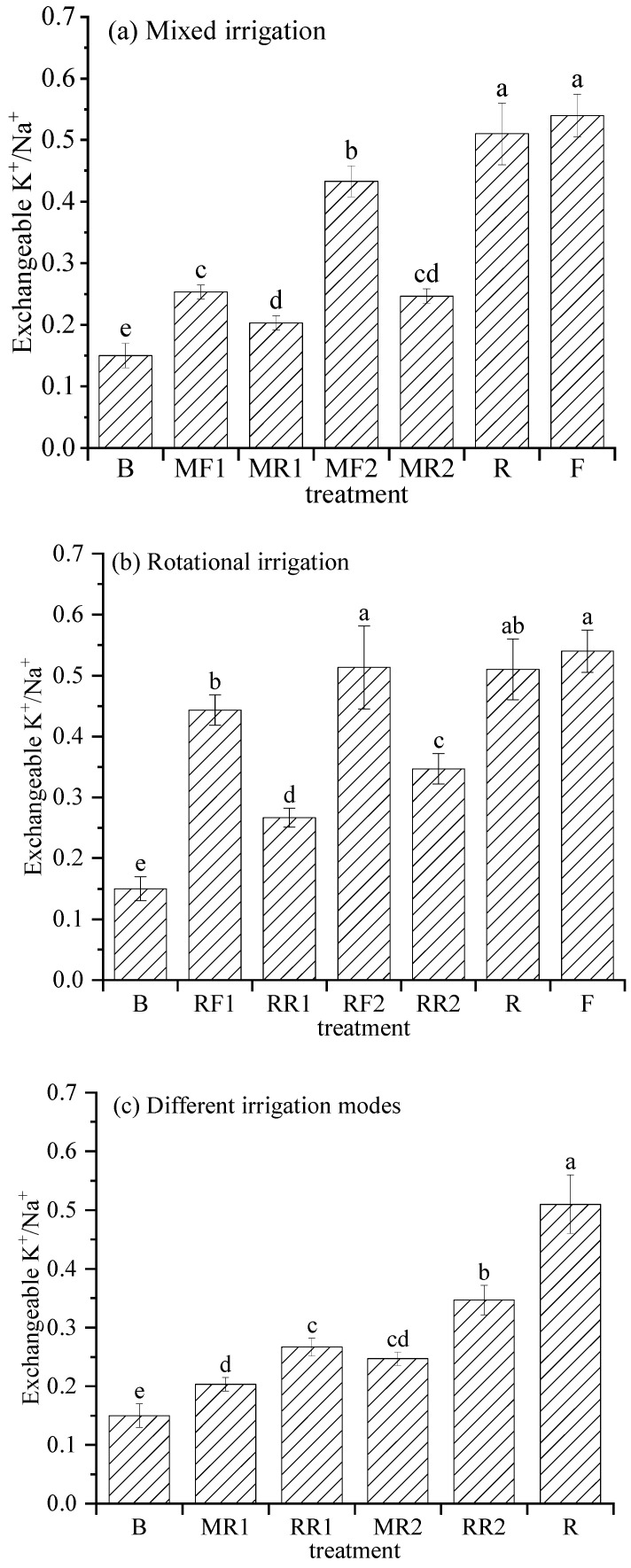
Variation in soil exchangeable K^+^/Na^+^ under different irrigation scenarios. Note: different lowercase letters on the points represent the significant differences at the level of 0.05.

**Figure 5 plants-11-02552-f005:**
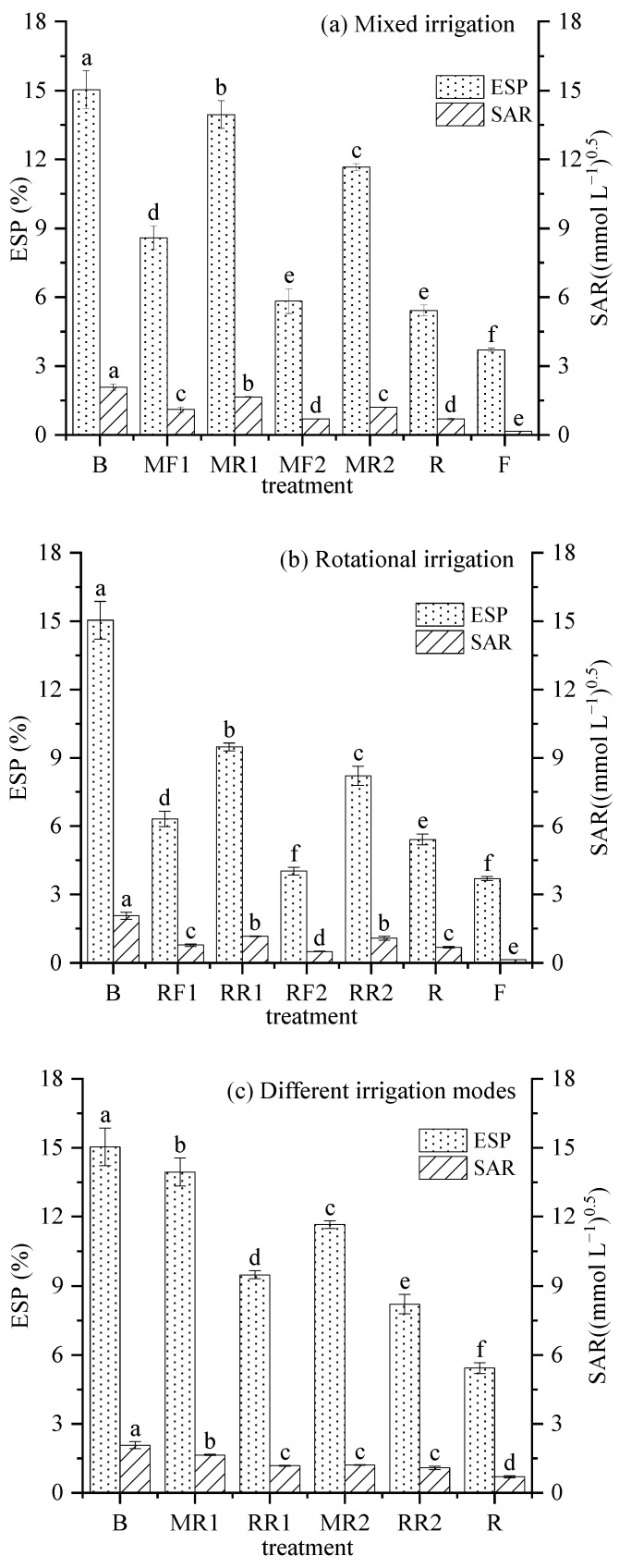
Variation in soil ESP and SAR after different irrigation scenarios. Note: different lowercase letters on the points represent the significant differences at the level of 0.05.

**Figure 6 plants-11-02552-f006:**
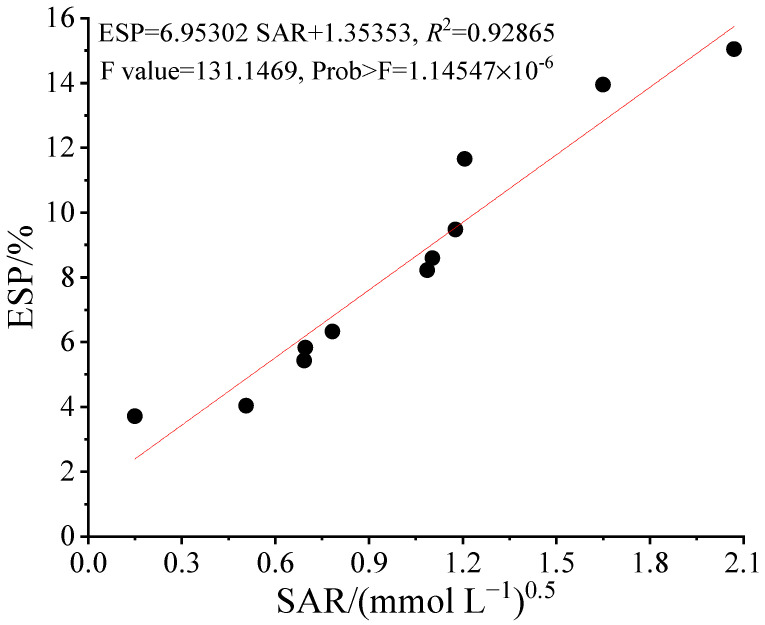
Relationship between soil ESP and SAR.

**Figure 7 plants-11-02552-f007:**
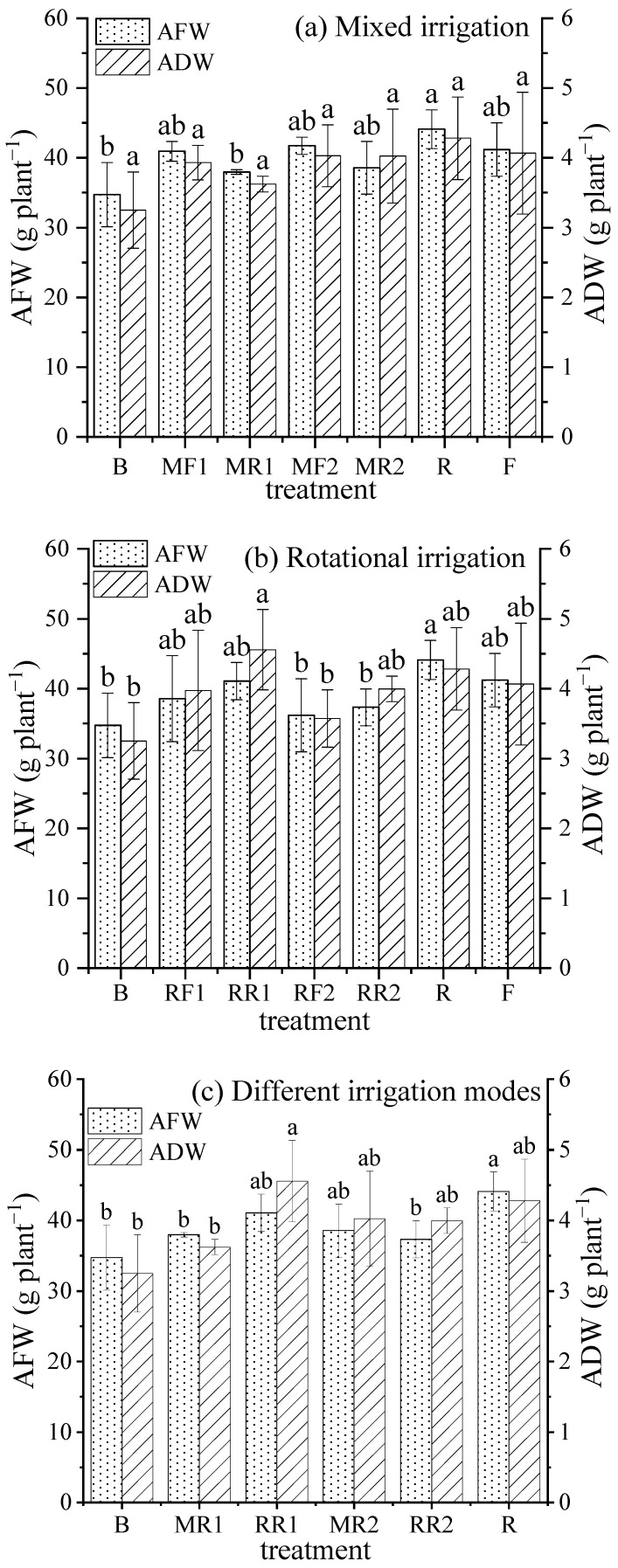
Variations in AFW and ADW under different irrigation scenarios. Note: different lowercase letters on the bars represent the significant differences at the level of 0.05.

**Table 1 plants-11-02552-t001:** Leaf sodium content under different irrigation modes.

Treatment	B	MR1	RR1	MR2	RR2	R
Na^+^ in leaf	15.07 ± 0.26a	14.46 ± 1.18a	12.36 ± 0.33b	12.91 ± 0.83b	12.43 ± 0.25b	10.02 ± 0.18c

Note: Leaf sodium content is in mg g^−1^; different lowercase letters behind the data represent the significant differences at the level of 0.05.

**Table 2 plants-11-02552-t002:** The established treatments of pot experiment for combined irrigation.

Treatment	F	B	MF1	MF2	R	MR1	MR2	RF1	RF2	RR1	RR2
Irrigation water	FW	BW	1:1 of BW to FW	1:2 of BW to FW	RW	1:1 of BW to RW	1:2 of BW to RW	FW-BW	FW-FW-BW	RW-BW	RW-RW-BW

Note: FW represents freshwater; BR represents brackish water of 3 g L^−1^; RW represents reclaimed water.

**Table 3 plants-11-02552-t003:** Quality of different water sources for the experiment.

Water Source	EC	pH	Na^+^	K^+^	HCO_3_^−^	Cl^−^	Ca^2+^	Mg^2+^	SO_4_^2−^	SAR	TN	TP	Pb	Cu	Zn	Cd
FW	0.321	8.31	0.4	0.04	1.96	0.85	0.98	0.61	1.08	0.34	1.17	0.02	ND	ND	ND	ND
RW	2.120	8.17	13.5	0.36	4.56	8.85	2.28	3.10	5.28	5.81	0.52	0.05	ND	ND	ND	ND
BW	6.100	8.41	57.8	0.05	2.32	54.20	1.08	0.71	0.96	43.21	1.31	0.02	ND	ND	ND	ND

Note: EC represents electrical conductivity, dS m^−1^; SAR represents sodium adsorption ratio, (mmol L^−1^)^0.5^; TN represents total nitrogen content, mg L^−1^; TP represents total phosphorus content, mg L^−1^; Pb, Cu, Zn, Cd, the unit is mg L^−1^; Na^+^, K^+^, HCO_3_^−^, Cl^−^, Ca^2+^, Mg^2+^, SO_4_^2−^, the unit is mmol L^−1^; ND indicates no detected: concentration were below the instrumental detection limit.

## Data Availability

The data used to support the findings of this study are available from the corresponding author upon request.

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
