# Peer review of "Does Short-Term Combined Irrigation Using Brackish-Reclaimed Water Cause the Risk of Soil Secondary Salinization?"

_plants, 2022, doi:10.3390/plants11192552_

Round 1

Reviewer 1 Report

Article title. The title shows a conclusion that with the information shown in the article is not demonstrated. A title that refers to the entire experiment would be more convenient.

Keywords: The words that appear in the title should not be used as keywords.

Units. The same format should be used throughout the article. Sometimes they appear as a fraction and sometimes as an exponent.

2. Materials and methods.

The irrigation section is poorly explained. It is not possible to reproduce the experiment exactly with the information that appears in the paper

2.1 Tested soil

Lines 118-119. ¿How has it been determined the field water-hold capacity of the soil?

2.2. Experimental device and scheme

Line 126. The duration of the experiment is very short. Drawing conclusions, even in the short term, is quite daring.

Lines 134-135. This text can be ignored, it contributes little: "It is known as the "king of vegetables" due to its delicious and nutritious."

Lines 152-154. Here it is indicated that the crop was irrigated every 4 days with 300 ml. Then it is indicated that the upper limit of the irrigation water was 75% of the water retention capacity of the field. What does this mean: 300 mL is 75% of the upper limit? Is crop consumption not taken into account?. This needs to be clarified.

Has drainage occurred? If so, it must have been quantified and analyzed. The irrigation section is poorly explained. It is not possible to reproduce the experiment exactly with the information that appears in the paper.

Table 1. In MT3, where 1:1 appears, maybe 1:2 should appear.

Table 2. In the same way that the quality of the mixtures of brackish water with recycled water appears, the quality of the mixtures of fresh water and brackish water should appear

Line 166. The units presentation must be unified.

2.3 Measured indexes and methods

Haven't measurements been made in the plant: production, biomass...? If so, the results should be displayed. This is very important as it can explain part of the results.

Measuring final soil moisture without information on water consumption by the crop or quantifying drainage does not allow for water balance

Lines 186-187. Missing line break before "2.4 Data analysis"

3. Results

3.1 Soil water and salts contents

Lines 195-198. The text that appears here fits more in the introduction section

Figure 1. In figures 1 (a) and 1 (b) the MT1 treatment does not show significant differences with respect to the MT2, MT3, RT1 and RT2 treatments in the soil moisture value. However, in Figure 1(c) there are significant differences. This is not very clear

Figure 1. What do the intervals shown in the graphs mean?

3.2 Soil exchangeable ions contents

Figure 2. What do the intervals shown in the graphs mean?

3.3 pH in soil

Figure 3. What do the intervals shown in the graphs mean?

Why is a different graph used in each section? If possible, the format of the graphs should be unified

Discussion

The differences in the ion content between the mixing treatments and the rotational treatments are explained because there are differences in the leaching or in the extraction of the plant, but no information is provided in this regard in the article.

Author Response

Article title. The title shows a conclusion that with the information shown in the article is not demonstrated. A title that refers to the entire experiment would be more convenient.

Reply: Thanks for your advice. We revised the title to “Does short-term combined irrigation using brackish-reclaimed water cause the risk of soil secondary salinization?”.

Keywords: The words that appear in the title should not be used as keywords.

Reply: Thanks. You are right. So, we change the keywords. Now the keywords are “mixed irrigation; rotational irrigation; exchangeable sodium percentage(ESP); sodium adsorption ratio(SAR); salinization”.

Units. The same format should be used throughout the article. Sometimes they appear as a fraction and sometimes as an exponent.

Reply: Thanks for your advice. We have checked and revised the units in order to keep the same format.

  1. Materials and methods.

The irrigation section is poorly explained. It is not possible to reproduce the experiment exactly with the information that appears in the paper

Reply: Thanks for your advice. We modified the irrigation section. Using the traditional surface irrigation methods, we irrigated the crop for 300 mL per pot(100% of the field water holding capacity) once the soil moisture was lower than 75% of the field water holding capacity.  The soil moisture of each pot was monitored by portable soil moisture meter. Due to the low air temperature and evaporation in the end of autumn, the crops were irrigated about every 4 days”.

2.1 Tested soil

Lines 118-119. ¿How has it been determined the field water-hold capacity of the soil?

Reply: Thanks for your advice. The field water-hold capacity was measured by other colleague. We used the data directly. The field water-hold capacity was determined by the cutting ring method. The undisturbed soil was collected from the experimental plot with cutting ring and brought back to the lab. Then, open the top cover of the cutting ring, put the cutting ring with the original soil sample (with the bottom cover with filter paper) into the container (the particle size of the sand is about 1mm, the thickness is more than 1cm, and a large layer of sand should be laid on top of the sand). Slowly pour clean water into the container until the water surface is 1~ 2 mm below the top of the cutting ring, soaking the soil sample of the cutting ring in water for no less than 24 hours to ensure that the soil sample is soaked evenly, so that the soil moisture in cutting ring is saturated. Then, place the soaked and saturated undisturbed soil on a cutting ring with air-dried soil covered with filter paper. After the air-dried soil absorbs the gravity water in the undisturbed soil (about 8 hours), the water content is determined by the drying method, which is the field water-holding capacity.

2.2. Experimental device and scheme

Line 126. The duration of the experiment is very short. Drawing conclusions, even in the short term, is quite daring.

Reply: Thanks for your advice. Due to the short reproductive period of Shanghai green, so the duration of the experiment is short. However, vegetable has a large water demand, so the irrigation frequency is high, compared to wheat or maize in the field. In future study, we will continue to the long-term or multi-season planting.

Lines 134-135. This text can be ignored, it contributes little: "It is known as the "king of vegetables" due to its delicious and nutritious."

Reply: Thanks. Okay, we deleted the text.

Lines 152-154. Here it is indicated that the crop was irrigated every 4 days with 300 ml. Then it is indicated that the upper limit of the irrigation water was 75% of the water retention capacity of the field. What does this mean: 300 mL is 75% of the upper limit? Is crop consumption not taken into account?. This needs to be clarified.

Has drainage occurred? If so, it must have been quantified and analyzed. The irrigation section is poorly explained. It is not possible to reproduce the experiment exactly with the information that appears in the paper.

Reply: Thanks. Very sorry for our carelessness. The lower limit is 75% of the field water-holding capacity , and the upper limit is 100% of the field water-holding capacity. During the process of experiment, no drainage occurred. The irrigation section has been rewritten.

Table 1. In MT3, where 1:1 appears, maybe 1:2 should appear.

Reply: Thanks. You are right. It should be 1:2, and we have modified it.

Table 2. In the same way that the quality of the mixtures of brackish water with recycled water appears, the quality of the mixtures of fresh water and brackish water should appear

Reply: Thanks. You are right. we are very sorry for our mistake. We emphasized on the mixture of brackish water with recycled water, and ignored the mixture of fresh water and brackish water. So, we did not measure the quality of the mixtures of fresh water and brackish water.

Line 166. The units presentation must be unified.

Reply: Thanks. The units have been unified.

2.3 Measured indexes and methods

Haven't measurements been made in the plant: production, biomass...? If so, the results should be displayed. This is very important as it can explain part of the results.

Measuring final soil moisture without information on water consumption by the crop or quantifying drainage does not allow for water balance

Reply: Thanks for your advice. No drainage occurred during the whole growth stages. We measured the data about fresh weight dry weight of aboveground. We have added the results about crop in the revised manuscript.

Lines 186-187. Missing line break before "2.4 Data analysis"

Reply: Thanks. This has been revised.

  1. Results

3.1 Soil water and salts contents

Lines 195-198. The text that appears here fits more in the introduction section

Figure 1. In figures 1 (a) and 1 (b) the MT1 treatment does not show significant differences with respect to the MT2, MT3, RT1 and RT2 treatments in the soil moisture value. However, in Figure 1(c) there are significant differences. This is not very clear

Reply: Thanks for your advice. We put the text to the introduction section. After checking, there were some errors in Figure1(c). We have replaced the picture. Thanks for your carefulness.

Figure 1. What do the intervals shown in the graphs mean?

Reply: Thanks. MC2,MC3,MT2,MC4,MT3,MT1,MC1, the order is convent to compared “ brackish-freshwater” mixed irrigation with “brackish-reclaimed water” mixed irrigation. Of course, the order is also that MC1,MC2,MC3, MC4, MT1, MT2,MT3. The same as follows in Figure 2 and Figure 3.

3.2 Soil exchangeable ions contents

Figure 2. What do the intervals shown in the graphs mean?

Reply: Thanks. MC2,MC3,MT2,MC4,MT3,MT1,MC1, the order is convent to compared “ brackish-freshwater” mixed irrigation with “brackish-reclaimed water” mixed irrigation. Of course, the order is also that MC1,MC2,MC3, MC4, MT1, MT2,MT3.

3.3 pH in soil

Figure 3. What do the intervals shown in the graphs mean?

Reply: Thanks. MC2,MC3,MT2,MC4,MT3,MT1,MC1, the order is convent to compared “ brackish-freshwater” mixed irrigation with “brackish-reclaimed water” mixed irrigation. Of course, the order is also that MC1,MC2,MC3, MC4, MT1, MT2,MT3.

Why is a different graph used in each section? If possible, the format of the graphs should be unified

Reply: Thanks. We use different graph in order to avoid a monotonous paper. In the revision, we unify the graphs according to your advice.

Discussion

The differences in the ion content between the mixing treatments and the rotational treatments are explained because there are differences in the leaching or in the extraction of the plant, but no information is provided in this regard in the article.

Reply: Thanks for your advice. Yes, you are right. in our manuscript there was no drainage. So, we rewrote the text. We measured Na+ content in plant, and added the data in revised manuscript.

Table 1  The Na+ content in plant under each treatments

Treatment

mean

std

MT2

14.46

1.18

MT3

12.91

0.83

RT1

12.36

0.33

RT2

12.43

0.25

Reviewer 2 Report

General remarks

The work studies the secondary risk of soil salinity with mixed irrigation and rotary irrigation with brackish water and reclaimed water or fresh water, in order to explore the safe and rational use of brackish water in areas where freshwater resources are scarce.

It is a very interesting topic and has applicability to identify water mixing strategies that allow saving fresh water without production losses, but the most important aspect that is production is not shown in this article.

It is also a current topic but the study focuses exclusively on the effects on the soil, without taking into account the crop, which, from my point of view, is outside the scope of the journal and for this reason it should be rejected.

In my opinion, the manuscript could be considered for this journal if the authors submit a new version including plant data.

The level of copying detected by the Turnitin program is very high (attachment) although most of it is collected from an article by the authors themselves, which would be duplication, not plagiarism, but I suggest that the authors correct this.

Even so, I have made a series of comments in case the authors want to take them into account to improve the article before sending it to a journal that covers the article's area of ​​specialization.

Specific remarks

In  Keywords section

Authors should remove words that are already in the title of the article, for example "salinization"

In  Materials and Methods

Line 115: “The tested soil was collected from topsoil in a field near the Agricultural Soil”, Please specify depth of profile sampled.

Line 143-144: “The specific experimental design is shown in Table 1”. Table 1 shows the established treatments, not the experimental design.

In Table 1. MT3 should be 1:2 of BW to RW not 1:1 of BW to RW. Please correct.

Line 152-153: “Due to the lower temperature and evaporation in the end of autumn, the crops were irrigated every 4 days with 300 mL (the upper limit of irrigation water was 75% of the field water-holding capacity)”. In a study of soil secondary salinity, it is very important to detail irrigation management. Please indicate the total volume of water supplied in the crop cycle, as well as detail if there was drainage. From the indicated data it can be deduced that the authors have maintained a constant irrigation throughout the cycle without considering the ETc of the crop.

Line 160-161: “The water qualities of different water sources are shown in Table 2”.The title of the table must be corrected and indicate that the characteristics of the mixtures are also mixed.

In Table 2: Table 2 shows the water qualities of different water sources and two mixtures, it would be necessary to indicate the resulting characteristics of the other two mixtures that are missing (MT2 and MT3). In addition, the title of the table must be corrected and indicate that the characteristics of the mixtures are also mixed.  It is also necessary to check the values of the mixtures, for example the Mg2+ in 1:2 of BW to RW, the value must be higher.

In  Results

In section 3.1 Soil water and salt contents. The determination of soil water content and salt contents cannot be properly discussed as the authors have not presented crop (production/biomass) or drainage data. The water content in the soil and salinity will be influenced by these parameters, both by the development of the crop and by irrigation management (drainage). In my opinion, it is necessary to add biomass/crop production data and whether or not there was drainage in order to correctly explain what happened in the soil.

In Figure 1. It is necessary to indicate what the interval of each of the bars means. It is necessary to do it in all the figures.

Other remarks

In addition, corrections text editing errors must be corrected, for example:

Unify units. In graphs dSm-1, in text dS/m

Author Response

General remarks

The work studies the secondary risk of soil salinity with mixed irrigation and rotary irrigation with brackish water and reclaimed water or fresh water, in order to explore the safe and rational use of brackish water in areas where freshwater resources are scarce.

It is a very interesting topic and has applicability to identify water mixing strategies that allow saving fresh water without production losses, but the most important aspect that is production is not shown in this article.

It is also a current topic but the study focuses exclusively on the effects on the soil, without taking into account the crop, which, from my point of view, is outside the scope of the journal and for this reason it should be rejected.

In my opinion, the manuscript could be considered for this journal if the authors submit a new version including plant data.

Reply: Thanks for your advice. We measured the data about fresh weight dry weight of aboveground. We have added the results about crop in the revised manuscript.

The level of copying detected by the Turnitin program is very high (attachment) although most of it is collected from an article by the authors themselves, which would be duplication, not plagiarism, but I suggest that the authors correct this.

Reply: Thanks for your advice. We restated the text in order to avoid the duplication.

Even so, I have made a series of comments in case the authors want to take them into account to improve the article before sending it to a journal that covers the article's area of ​​specialization.

Specific remarks

In Keywords section

Authors should remove words that are already in the title of the article, for example "salinization"

Reply: Thanks for your advice. We have revised the all keywords.

In Materials and Methods

Line 115: “The tested soil was collected from topsoil in a field near the Agricultural Soil”, Please specify depth of profile sampled.

Reply: Okay, the tested soil was collected from topsoil (0-20 cm) in a field .

Line 143-144: “The specific experimental design is shown in Table 1”. Table 1 shows the established treatments, not the experimental design.

Reply: yes, you are right. we revised the title of table1. The title is “The established treatments of pot experiment for combined irrigation”.

In Table 1. MT3 should be 1:2 of BW to RW not 1:1 of BW to RW. Please correct.

Reply: Thanks for your advice. We have corrected the error. Thanks for your carelessness again.

Line 152-153: “Due to the lower temperature and evaporation in the end of autumn, the crops were irrigated every 4 days with 300 mL (the upper limit of irrigation water was 75% of the field water-holding capacity)”. In a study of soil secondary salinity, it is very important to detail irrigation management. Please indicate the total volume of water supplied in the crop cycle, as well as detail if there was drainage. From the indicated data it can be deduced that the authors have maintained a constant irrigation throughout the cycle without considering the ETc of the crop.

Reply: Thanks for your advice. We are sorry for the confusing description. We modified the irrigation section. During the experiment, no drainage occurred and the bottom of the pot have several holes. Using the traditional surface irrigation methods, we started to irrigate crop for 300 mL per pot (100% of the field water holding capacity) when soil moisture was lower than 75% of the field water holding capacity. One pot was selected to monitor the soil moisture by portable soil moisture meter. Due to the lower temperature and evaporation in the end of autumn, the crops were irrigated every about 4 days.

Line 160-161: “The water qualities of different water sources are shown in Table 2”.The title of the table must be corrected and indicate that the characteristics of the mixtures are also mixed.

In Table 2: Table 2 shows the water qualities of different water sources and two mixtures, it would be necessary to indicate the resulting characteristics of the other two mixtures that are missing (MT2 and MT3). In addition, the title of the table must be corrected and indicate that the characteristics of the mixtures are also mixed.  It is also necessary to check the values of the mixtures, for example the Mg2+ in 1:2 of BW to RW, the value must be higher.

Reply: Thanks. You are right. we are very sorry for our mistake. The study on the mixture of fresh water and brackish water is too many, and we emphasized  the mixture of brackish water with recycled water, and took the mixture of fresh water and brackish water as the control to analyze the difference between them. So, we did not measure the quality of the mixtures of fresh water and brackish water. We checked the value, and it is right. We also noticed the point during our experiment, and it has been retested.  The title of Table has been corrected.

In Results

In section 3.1 Soil water and salt contents. The determination of soil water content and salt contents cannot be properly discussed as the authors have not presented crop (production/biomass) or drainage data. The water content in the soil and salinity will be influenced by these parameters, both by the development of the crop and by irrigation management (drainage). In my opinion, it is necessary to add biomass/crop production data and whether or not there was drainage in order to correctly explain what happened in the soil.

Reply: Thanks for your advice. Yes, you are right. In our manuscript there was no drainage. We added the aboveground biomass in revised manuscript.

In Figure 1. It is necessary to indicate what the interval of each of the bars means. It is necessary to do it in all the figures.

Reply: Thanks. MC2,MC3,MT2,MC4,MT3,MT1,MC1, the order is convent to compared “ brackish-freshwater” mixed irrigation with “brackish-reclaimed water” mixed irrigation. Of course, the order is also that MC1,MC2,MC3, MC4, MT1, MT2,MT3.

Other remarks

In addition, corrections text editing errors must be corrected, for example: Unify units. In graphs dSm-1, in text dS/m

Reply: Thanks for your advice. We have checked and revised the units in order to keep the same format.

Reviewer 3 Report

The authors provided a interesting study on a very hot topic: the need to find alternative water sources to cope with water scarcity.

The MS is well structured and the experiment is well described.

However, one of the main questions is not answered: one of the main questions is not answered: what is the impact of using brackish or reclaimed water on crop yield?
Especially as it is a crop consumed fresh.

Also, the title should be rethought... the title shouldn't be a statement.   

Author Response

The authors provided a interesting study on a very hot topic: the need to find alternative water sources to cope with water scarcity.

The MS is well structured and the experiment is well described.

However, one of the main questions is not answered: one of the main questions is not answered: what is the impact of using brackish or reclaimed water on crop yield? Especially as it is a crop consumed fresh.

Also, the title should be rethought... the title shouldn't be a statement.   

Reply: Thanks for your advice. We revised the title to “Does short-term combined irrigation using brackish-reclaimed water cause the risk of soil secondary salinization?”. We measured the data about fresh weight dry weight of aboveground. We have added the results about crop in the revised manuscript.

Reviewer 4 Report

I find the paper interesting and containing elements of a scientific novelty. I only recommend to write shortly about the reference of the performed research to the current scientific works ( state of the art) in the discussion section around lines 433 - 444. How are your results relevant to the achievements of other works? Would you then refer to this also in the conclusions section, how you could go beyond the current state of the knowledge?

Author Response

I find the paper interesting and containing elements of a scientific novelty. I only recommend to write shortly about the reference of the performed research to the current scientific works ( state of the art) in the discussion section around lines 433 - 444. How are your results relevant to the achievements of other works? Would you then refer to this also in the conclusions section, how you could go beyond the current state of the knowledge?

Reply: Thanks for your advice. Now, few study about combined utilization of reclaimed water and brackish is reported. Our research could provide the new perspectives to solve the existed problems of brackish water or reclaimed water. Our research is a continuation for single utilization of reclaimed or brackish water. However, there are still much scientific issues need to study, such as the effects of long-term irrigation on soil, crop and soil microorganisms, as well as the interaction mechanism of ions.

Round 2

Reviewer 1 Report

Comments

2. Materials and methods

Lines 173-178. The English language of these lines must be corrected.

3. Results

3.7 Crop biomass

Review the full text of the section. It is too convoluted. Avoid comments such as: "...it tends to increase..." or "...it tends to decrease..." if there are no significant differences between treatments.

When the rotational treatments are compared with the mixture ones, there are only differences in ADW between RT1 and MT2

Discussion

Lines 571-574.

The leaf sodium content data reflected in the discussion must previously appear in the results section.

Suggestions

Could be used nomenclatures of the treatments that facilitate the reading and understanding of the article. For example, 1:1 and 1:2 mixtures of fresh and brackish water and reclaimed and brackish water could be named: MF1, MF2 and MR1,MR2 respectively. In the same way the rotations could be called: RF1, RF2 and RR1, RR2

Author Response

Comments

  1. Materials and methods

Lines 173-178. The English language of these lines must be corrected.

Reply: Thanks for your advice. We corrected the English language in line 173-178.

  1. Results

3.7 Crop biomass

Review the full text of the section. It is too convoluted. Avoid comments such as: "...it tends to increase..." or "...it tends to decrease..." if there are no significant differences between treatments.

When the rotational treatments are compared with the mixture ones, there are only differences in ADW between RT1 and MT2

Reply: thanks for your advice. We checked and revised the text.

Discussion

Lines 571-574.

The leaf sodium content data reflected in the discussion must previously appear in the results section.

Reply: thanks for your advice. We added the leaf sodium content data in the results section.

Suggestions

Could be used nomenclatures of the treatments that facilitate the reading and understanding of the article. For example, 1:1 and 1:2 mixtures of fresh and brackish water and reclaimed and brackish water could be named: MF1, MF2 and MR1,MR2 respectively. In the same way the rotations could be called: RF1, RF2 and RR1, RR2

Reply: thanks for your suggestions. We revised the name of treatment according to your advice.

Reviewer 2 Report

General remarks

The work studies the secondary risk of soil salinity with mixed irrigation and rotary irrigation with brackish water and reclaimed water or fresh water, in order to explore the safe and rational use of brackish water in areas where freshwater resources are scarce.

It is a very interesting topic and has applicability to identify water mixing strategies that allow saving fresh water without production losses, but the most important aspect that is production is not shown in this article.

It is also a current topic but the study focuses exclusively on the effects on the soil, without taking into account the crop, which, from my point of view, is outside the scope of the journal and for this reason it should be rejected.

In my opinion, the manuscript could be considered for this journal if the authors submit a new version including plant data.

Reply: Thanks for your advice. We measured the data about fresh weight dry weight of aboveground. We have added the results about crop in the revised manuscript.

Reply to reply: Perfect, the study is now more complete. Thanks.

The level of copying detected by the Turnitin program is very high (attachment) although most of it is collected from an article by the authors themselves, which would be duplication, not plagiarism, but I suggest that the authors correct this.

Reply: Thanks for your advice. We restated the text in order to avoid the duplication.

Reply to reply: Perfect. Thanks.

Even so, I have made a series of comments in case the authors want to take them into account to improve the article before sending it to a journal that covers the article's area of ​​specialization.

Specific remarks

In Keywords section

Authors should remove words that are already in the title of the article, for example "salinization"

Reply: Thanks for your advice. We have revised the all keywords.

Reply to reply: Perfect. Thanks.

In Materials and Methods

Line 115: “The tested soil was collected from topsoil in a field near the Agricultural Soil”, Please specify depth of profile sampled.

Reply: Okay, the tested soil was collected from topsoil (0-20 cm) in a field .

Reply to reply: Perfect. Thanks.

Line 143-144: “The specific experimental design is shown in Table 1”. Table 1 shows the established treatments, not the experimental design.

Reply: yes, you are right. we revised the title of table1. The title is “The established treatments of pot experiment for combined irrigation”.

Reply to reply: Perfect. Thanks.

In Table 1. MT3 should be 1:2 of BW to RW not 1:1 of BW to RW. Please correct.

Reply: Thanks for your advice. We have corrected the error. Thanks for your carelessness again.

Reply to reply: Perfect. Thanks.

Line 152-153: “Due to the lower temperature and evaporation in the end of autumn, the crops were irrigated every 4 days with 300 mL (the upper limit of irrigation water was 75% of the field water-holding capacity)”. In a study of soil secondary salinity, it is very important to detail irrigation management. Please indicate the total volume of water supplied in the crop cycle, as well as detail if there was drainage. From the indicated data it can be deduced that the authors have maintained a constant irrigation throughout the cycle without considering the ETc of the crop.

Reply: Thanks for your advice. We are sorry for the confusing description. We modified the irrigation section. During the experiment, no drainage occurred and the bottom of the pot have several holes. Using the traditional surface irrigation methods, we started to irrigate crop for 300 mL per pot (100% of the field water holding capacity) when soil moisture was lower than 75% of the field water holding capacity. One pot was selected to monitor the soil moisture by portable soil moisture meter. Due to the lower temperature and evaporation in the end of autumn, the crops were irrigated every about 4 days.

Reply to reply: You continue without indicating the total volume of irrigation in the cycle. It is an important piece of information because it allows calculating the amount of salts that have been added to the soil in each treatment.

Line 160-161: “The water qualities of different water sources are shown in Table 2”.The title of the table must be corrected and indicate that the characteristics of the mixtures are also mixed.

In Table 2: Table 2 shows the water qualities of different water sources and two mixtures, it would be necessary to indicate the resulting characteristics of the other two mixtures that are missing (MT2 and MT3). In addition, the title of the table must be corrected and indicate that the characteristics of the mixtures are also mixed. It is also necessary to check the values of the mixtures, for example the Mg2+ in 1:2 of BW to RW, the value must be higher.

Reply: Thanks. You are right. we are very sorry for our mistake. The study on the mixture of fresh water and brackish water is too many, and we emphasized the mixture of brackish water with recycled water, and took the mixture of fresh water and brackish water as the control to analyze the difference between them. So, we did not measure the quality of the mixtures of fresh water and brackish water. We checked the value, and it is right. We also noticed the point during our experiment, and it has been retested. The title of Table has been corrected.

Reply to reply: I am sorry to insist that there must be an error in the qualities of water resulting from the mixture, if not, how is it possible that the mixtures have a lower pH than the mixed waters?, and how is it possible that they have higher bicarbonates? The resulting water values should be close to this attached table, since I have calculated the weighted mean in each case.

EC

pH

Na+

K+

HCO3-

Cl-

Ca2+

Mg2+

SO42-

RW

2.12

8.17

13.50

0.36

4.56

8.85

2.28

3.10

5.28

BW

6.10

8.41

57.80

0.05

2.32

54.20

1.08

0.71

0.96

1:1 BW to RW

4.11

8.29

35.65

0.21

3.44

31.53

1.68

1.91

3.12

1:2 BW to RW

3.45

8.25

28.27

0.26

3.81

23.97

1.88

2.30

3.84

In Results

In section 3.1 Soil water and salt contents. The determination of soil water content and salt contents cannot be properly discussed as the authors have not presented crop (production/biomass) or drainage data. The water content in the soil and salinity will be influenced by these parameters, both by the development of the crop and by irrigation management (drainage). In my opinion, it is necessary to add biomass/crop production data and whether or not there was drainage in order to correctly explain what happened in the soil.

Reply: Thanks for your advice. Yes, you are right. In our manuscript there was no drainage. We added the aboveground biomass in revised manuscript.

Reply to reply: Perfect. Thanks.

In Figure 1. It is necessary to indicate what the interval of each of the bars means. It is necessary to do it in all the figures.

Reply: Thanks. MC2,MC3,MT2,MC4,MT3,MT1,MC1, the order is convent to compared “ brackish-freshwater” mixed irrigation with “brackish-reclaimed water” mixed irrigation. Of course, the order is also that MC1,MC2,MC3, MC4, MT1, MT2,MT3.

Reply to reply: Excuse me, it is necessary that you indicate in all the figures the meaning of the error bars shown. Is it the standard error or standar desviation? Is it the range of variation? Is it the typical deviation? You specify the meaning of the letters but not of the error bars.

Other remarks

In addition, corrections text editing errors must be corrected, for example: Unify units. In graphs dSm-1, in text dS/m

Reply: Thanks for your advice. We have checked and revised the units in order to keep the same format.

Reply to reply: Perfect. Thanks.

New comments:

In Table 2, Are the SAR units correct? Please correct.

In Figure 7 (a) there must be an error in the letters that indicate statistical differences in ADW, with the current letters (a and ab) there would be no significant differences between treatments.

In the conclusions section it would be convenient to introduce a conclusion of the results obtained in the plant.

Author Response

You continue without indicating the total volume of irrigation in the cycle. It is an important piece of information because it allows calculating the amount of salts that have been added to the soil in each treatment.

Reply: We added the irrigation times and the total volume after the irrigation treatments.

New comments:

In Table 2, Are the SAR units correct? Please correct.

Reply: thanks for your advice. We checked the units. In our manuscript, the unit of SAR is (mmol L-1)0.5. and its calculated equation is as follows:

                 equation(1)

Where Na+, Ca2+ and Mg2+ concentrations at a soil-to-water ratio of 1:5 are in mmol L-1.

In addition, considering your advice and scientific rigor, we delete the data about mixture and maintain the data of water sources.

In Figure 7 (a) there must be an error in the letters that indicate statistical differences in ADW, with the current letters (a and ab) there would be no significant differences between treatments.

Reply: thanks for your suggestions. Yes, you are right. We have checked and revised the figure.

In the conclusions section it would be convenient to introduce a conclusion of the results obtained in the plant.

Reply: thanks for your suggestions. We added a conclusion about plant in the conclusions section.